# Controversies in Endoscopic Ultrasound-Guided Biliary Drainage

**DOI:** 10.3390/cancers16091616

**Published:** 2024-04-23

**Authors:** Christoph Frank Dietrich, Paolo Giorgio Arcidiacono, Manoop S. Bhutani, Barbara Braden, Eike Burmester, Pietro Fusaroli, Michael Hocke, Andrè Ignee, Christian Jenssen, Abed Al-Lehibi, Emad Aljahdli, Bertrand Napoléon, Mihai Rimbas, Giuseppe Vanella

**Affiliations:** 1Department Allgemeine Innere Medizin der Kliniken (DAIM) Hirslanden Beau Site, Salem und Permanence, 3013 Bern, Switzerland; 2Division of Pancreatobiliary Endoscopy and Endosonography, IRCCS San Raffaele Scientific Institute, Vita-Salute San Raffaele University, 20132 Milan, Italy; arcidiacono.paologiorgio@hsr.it (P.G.A.); vanella.giuseppe@hsr.it (G.V.); 3Department of Gastroenterology Hepatology and Nutrition, The University of Texas MD Anderson Cancer Center, Houston, TX 77030, USA; manoop.bhutani@mdanderson.org; 4Medical Department B, University Münster, Albert-Schweitzer-Campus 1, 48149 Münster, Germany; barbara.braden@ukmuenster.de; 5Medizinische Klinik I, Sana Kliniken Luebeck, 23560 Luebeck, Germany; burmester.buc@t-online.de; 6Department of Medical and Surgical Sciences, Gastrointestinal Unit, University of Bologna/Hospital of Imola, 40126 Bologna, Italy; 7Medical Department II, Helios Klinikum Meiningen, 98617 Meiningen, Germany; michaelhocke1@aol.com; 8Klinikum Würzburg Mitte, Standort Juliusspital, 97074 Würzburg, Germany; andre.ignee@kwm-klinikum.de; 9Medical Department, Krankenhaus Maerkisch-Oderland, 15441 Strausberg and Brandenburg Institute of Clinical Ultrasound at Medical University Brandenburg, 16816 Neuruppin, Germany; c.jenssen@khmol.de; 10Gastroenterology & Hepatology Department, King Fahad Medical City, Riyadh 11525, Saudi Arabia; aallehibi@kfmc.med.sa; 11Faculty of Medicine, King Abdulaziz University, Gastrointestinal Oncology Unit, King Abdul-Aziz University Hospital (KAUH), Jeddah 22252, Saudi Arabia; ealjahdli@gmail.com; 12Hopital Privé J Mermoz Ramsay Générale de Santé, 69008 Lyon, France; dr.napoleon@wanadoo.fr; 13Department of Gastroenterology, Clinic of Internal Medicine, Colentina Clinical Hospital, Carol Davila University of Medicine, 050474 Bucharest, Romania; mrimbas@gmail.com

**Keywords:** acute cholecystitis, endoscopic retrograde cholangiopancreatography, endoscopic ultrasound, endoscopic, ultrasound-guided biliary drainage, endoscopic ultrasound-guided gallbladder drainage, obstructive jaundice

## Abstract

**Simple Summary:**

In this review, first, the history of EUS-guided biliary drainage (EUS-BD) is summarized. In the following chapters controversies arising from various approaches and challenges in EUS-BD, EUS-guided gallbladder drainage (EUS-GBD) and alternatively performed procedures are discussed from different points of view on the background of the available evidence. In addition, for all topics arguments in favor and against the techniques are described and reflected. The topics include “Why do we need procedures other than ERCP?”; “Should EUS-BD and ERCP be performed by the same operator?”; rendezvous techniques, including “Should rendezvous be used first?” and “Which rendezvous route should be used?”; percutaneous transhepatic cholangiography and biliary drainage (PTBD); “Should PTBD and EUS-BD be performed by the same physician?”; “Do we need cystotomes?”; “Do we need bougies?”; “Are all EUS needles the same for EUS-BD?”; “Plastic or metal stents?”; and adverse events.

**Abstract:**

In this 14th document in a series of papers entitled “*Controversies in Endoscopic Ultrasound*” we discuss various aspects of EUS-guided biliary drainage that are debated in the literature and in practice. Endoscopic retrograde cholangiography is still the reference technique for therapeutic biliary access, but EUS-guided techniques for biliary access and drainage have developed into safe and highly effective alternative options. However, EUS-guided biliary drainage techniques are technically demanding procedures for which few training models are currently available. Different access routes require modifications to the basic technique and specific instruments. In experienced hands, percutaneous transhepatic cholangiodrainage is also a good alternative. Therefore, in this paper, we compare arguments for different options of biliary drainage and different technical modifications.

## 1. Introduction 

In this review, controversies arising from various approaches and challenges in EUS-guided biliary drainage are discussed from different points of view based on the background of the available evidence.

## 2. History of EUS-Guided Biliary Drainage 

Historically, the possibility of EUS-guided biliary drainage (EUS-BD) was initiated by a publication of Wiersema et al. in 1996. They reported the outcome of EUS-guided cholangiopancreaticography in 11 out of 205 patients in whom ERCP was not possible: in these 11 patients, diagnostic EUS-guided transduodenal cholangiography (*n* = 10) or a transgastric pancreaticography (*n* = 1) was successfully performed in 8 patients using a linear echoendoscope [1]. Five years later, in 2001, Giovannini et al. reported the first case of a therapeutic EUS-guided biliary transduodenal drainage in a patient with a bile duct obstruction due to a pancreatic mass. In this case, a two-step technique was used: initially the authors inserted a linear EUS probe and punctured the common bile duct (CBD) transduodenally; after contrast injection and the insertion of a 0.035 inch guidewire, the EUS-instrument was removed, and a duodenoscope was inserted. Finally, a plastic stent was successfully placed over the guidewire into the CBD [2].

Burmester et al. first published in 2003 the direct one-step approach. They reported four cases in which the simultaneous puncture and insertion of plastic stents were performed over a linear echoendoscope into the CBD and left hepatic duct as antegrade choledochoduodenostomy (CDS). The procedures served as retrograde hepatico-jejunostomy (HJS) and hepaticogastrostomy (HGS), respectively, over the left intrahepatic duct in two operated patients (gastrectomy with Roux-en-Y anastomosis; BII resection) [3]. In 2004, Mallery and colleagues first reported a successful rendezvous drainage maneuver in patients with biliary and pancreatic obstruction. They performed an EUS-guided transgastric/transduodenal puncture and placed a guidewire through either the CBD or the pancreatic duct into the duodenum. Finally, they successfully placed a stent in three out of six cases using ERCP as a rendezvous maneuver [4]. 

These innovative techniques created widespread interest in EUS-guided bile duct drainage. Subsequently, increasing numbers of predominantly small retrospective case series, single or multicentric experiences, and uncontrolled studies were published with technical and clinical success rates reaching 69–100% and 70–100%, respectively [5]. Initially, the main indication was an unsuccessful ERCP in advanced tumors with the infiltration of the papilla, the duodenum, the CBD and the hepatic bifurcation and in anatomical variants (e.g., duodenal diverticulum) or with surgically altered anatomy (Billroth’s II resection, gastrectomy or pancreaticoduodenectomy with Roux-en-Y reconstruction, hepaticojejunostomy). Subsequently, EUS-guided maneuvers were applied also to patients with benign diseases in whom a previous cannulation of the CBD had failed due to duodenal diverticulum, choledochal cysts, chronic pancreatitis or postsurgical strictures. In these cases, successful EUS-guided biliary accesses, especially using a rendezvous technique, were reported. A significantly higher success rate of EUS-guided procedures was observed in malignant diseases compared to benign conditions (90.2% vs. 77.3%; *p* = 0.02) [6,7,8,9,10]. In 2011, an expert consortium suggested that EUS-guided biliary drainage is generally indicated if biliary drainage is necessary and ERCP failed or was not feasible due to surgically altered anatomy of the upper gastrointestinal tract, gastric or duodenal obstruction, the non-traversable obstruction of the papilla or CBD or the presence of anatomical variants (e.g., duodenal diverticulum) [11]. 

In 2016, the EFSUMB guidelines on Interventional Ultrasound (INVUS) recommended EUS-BD in patients with malignant obstructive jaundice and failed ERCP as an alternative to percutaneous transhepatic biliary drainage (PTBD) and/or surgery [12]. This recommendation was based on some retrospective studies and one small randomized controlled trial comparing EUS-BD and PTBD [13,14,15]. This recommendation was subsequently supported by several retrospective and prospective studies (10 studies, 545 patients overall) with different techniques (PTBD vs. EUS-BD, including choledochoduodenostomy, EUS-guided rendezvous, EUS-guided gallbladder drainage). The technical and clinical success rates for EUS-BD and PTBD were, respectively, 90.8% vs. 90.3% and 89.1% vs. 73%. Of note, a significant higher rate of adverse events and re-intervention was observed in the PTBD group [16]. 

In 2012, Dhir et al. performed the EUS-guided rendezvous technique to obtain biliary access in 58 patients in whom selective cannulation had failed after five attempts using a sphincterotome and guidewire. The results were compared with those of a historical cohort of 144 patients who underwent precut sphincterotomy. Technical success was significantly higher for EUS-guided rendezvous access than for precut sphincterotomy (98.3% vs. 90.3%; *p* = 0.03). The rates of procedural complications did not differ significantly between both biliary access techniques (3.4% vs. 6.9%, *p* = 0.27) [6].

In 2016–2018, four prospective, single- and multicenter studies with 304 patients were published. The technical and clinical success rates for EUS and ERCP were 93.4% vs. 95.6% and 95.7% vs. 92.8%, respectively. In comparison with ERCP, EUS had a significantly lower rate of pancreatitis (0% vs. 19.7–35.7%) [16]. However, the technique of EUS-guided BD was not standardized, and no pooled data were available comparing the efficacy of different devices [17]. The best individual approach was based mainly on the fact whether the papilla was accessible or not. In 2018, an expert group recommended the rendezvous technique whenever possible, especially in benign diseases, followed by CDS and finally HGS/HJS, both in the antegrade or retrograde manner [18]. Other groups developed an algorithm for the initial access to the CBD based on the patient’s anatomy or enhanced guidewire manipulation for EUS-BD after failed ERCP [8,19,20]. In 2015, Artifon et al. compared the outcomes of EUS-HGS and EUS-CDS in a prospective, randomized controlled trial including 49 patients with distal malignant biliary obstruction. The technical and clinical success rates for HGS and CDS were 96% vs. 91% and 91% vs. 77%, respectively with no significant difference between the techniques. The immediate adverse event rates were similar with 20% in HGS and 12.5% in CDS [21].

Khashab et al. reported similar results for the placement of stents in an international multicenter comparative trial including 121 patients who underwent CDS (*n* = 60) or HGS [22] after failed ERCP. However, CDS was associated with a significant shorter hospital stay (5.6 ± 6 days vs. 12.7 ± 11.5 days, *p* < 0.001) and fewer procedural and stent-related complications like occlusion and migration (13.3% vs. 26.2%) [23]. In 2019, Hedjoudje et al. confirmed these data in a meta-analysis with no differences in the technical and clinical success rates of CDS and HGS, including 17 studies with 686 patients, although the complication rate was higher in the HGS group (OR CDS 1.09 vs. HGS 2.07; *p* < 0.001), especially due to stent dysfunction [24]. 

Recent randomized controlled multicenter studies have shown non-inferiority when EUS-BD was performed for distal malignant bile duct obstruction compared to ERCP as the first-choice endoscopic drainage procedure [25,26,27,28]. Recently, the European Society of Gastrointestinal Endoscopy (ESGE) published a technical review of therapeutic EUS with detailed technical recommendations for EUS-BD. The placement of either partially or fully covered self-expandable metal stents (SEMSs) in EUS-HGS and either SEMSs or small-caliber lumen-apposing metal stents (LAMSs) in EUS-CD were strongly recommended [29].

## 3. Why Do We Need Procedures Other Than ERCP? 

ERCP was introduced in the late 1960s [30] as a diagnostic tool for the bilio-pancreatic tree. Since then, less invasive imaging techniques such as magnetic resonance cholangiopancreatography and EUS have largely replaced its diagnostic role. Endoscopic sphincterotomy facilitates access to the CBD for stone extraction and stenting, shifting the role of ERCP to therapeutic only. Until now, ERCP has been considered worldwide as a first line treatment in the clinical management of biliary obstruction caused by either benign or malignant pathology. Endoscopic transpapillary stenting via ERCP enables biliary drainage with a high success rate (90–95%) but is associated with considerable adverse event rates. Acute post-ERCP pancreatitis, perforation, bleeding, cholangitis or stent dysfunction that may require re-interventions may occur in 1 out of 15–20 patients [31]. Moreover, selective bile duct cannulation from the papilla for therapeutic biliary intervention cannot be achieved in approximately 10% of patients [31,32]. Furthermore, in patients with a complex duodenal diverticulum, surgically altered anatomy or duodenal obstruction, endoscopic access to the papilla may be technically impossible. In using all available techniques in patients with normal anatomy and native papilla, CBD cannulation should be achievable in at least 90% of cases [33,34]. As the endoscopic transpapillary approach for biliary stenting in patients with malignant biliary, obstruction is unsuccessful in 5–10% of cases and is associated with a significant adverse event rate of about 5%; alternative techniques for biliary drainage are needed that provide a high success rate and a good safety profile. 

### 3.1. Arguments in Favor of EUS-BD

#### 3.1.1. EUS-BD of the Bile Duct

In patients with different diseases including biliary strictures, where attempts at ERCP have failed, EUS biliary drainage is an adequate alternative to the PTBD [35,36]. The higher morbidity of PTBD was never undoubtfully proven in prospective studies. PTBD is frequently performed as a rescue maneuver in several centers. EUS-BD can be conducted in patients with ascites, which is a relative contraindication to PTBD [37]. In cases of failed cannulation in surgically altered anatomy, balloon enteroscopy-assisted ERCP is another alternative in specialized centers (10–12), even if initial comparative data have demonstrated significantly lower technical and clinical success rates and an increased procedure duration compared to EUS-BD [38], which has conversely shown a slightly increased rate of predominantly mild/moderate adverse events. 

EUS-BD can be performed by the same endoscopist who attempted ERCP, without the need to reschedule another procedure or refer the patient to an interventional radiologist, with a potential reduction in hospital stay. Furthermore, PTBD usually requires external drainage, causing discomfort and pain to the patient. Even in specialized centers where subsequent internalization to metal stents is the standard of care, this will require a repeated procedure and may actually be impossible in a significant proportion of patients, for example, in cases of unnegotiable stenosis or where previously placed biliary stents impede further maneuvers. 

EUS-BD has the potential to compete with ERCP as a primary drainage procedure in patients with malignant biliary obstruction in non-surgical candidates. Furthermore, in recent series of patients who underwent subsequent surgical resection after EUS BD there was no increase in the surgical complication rate in the short-term follow-up, albeit a longer follow-up period is still required to assess oncological outcomes [39,40] [Figure 1, Figure 2, Figure 3 and Figure 4].

#### 3.1.2. EUS-GBD

In patients presenting with acute cholecystitis who have a high surgical risk, EUS-GBD may avoid the high surgical complication rate [41] and eliminate the discomfort caused by the external drainage tubes, which, in 8.6% of patients, are accidentally dislodged, resulting in the need for repeat procedures [41]. In the only international randomized control trial on this topic, EUS-GBD significantly reduced postprocedural pain, adverse events, re-interventions and recurrent cholecystitis, provided that the intervention was performed by experts. Further studies have demonstrated the long-term efficacy of EUS-GBD [42,43]. A recent meta-analysis found similar early adverse advents of EUS-GBD using LAMS compared to percutaneous gallbladder drainage, but EUS-GBD using LAMS was associated with shorter hospital stays and a significantly lower rate of delayed (OR: 0.21) and overall adverse events (OR: 0.43) [44].

### 3.2. Arguments against EUS-Guided Drainage

The complexity and the low case load of EUS-BD procedures requires a high degree of expertise both for EUS and ERCP, which is uncommonly found outside tertiary care centers, thereby limiting their widespread use. Even in expert centers, the EUS drainage of the biliary tree presents several technical challenges, especially if the transhepatic access is used. The target structure of the bile ducts has a diameter of some millimeters and is moving. The puncture window is small, limited by the need to avoid a transmediastinal approach and interposing vessels. A small contrast or air extravasation could hinder further attempts to target the same duct, although it might still admit interventions in other accessible ducts.

In elderly patients who undergo ERCP, the most common concomitant chronic diseases are of cardiovascular and cerebrovascular aitiology, often requiring anticoagulation or antiplatelet therapy. This might increase the potential bleeding risk for EUS-guided drainage, which is considered a high-risk procedure [45].

In addition, there is a lack of training models for this extremely difficult procedure that could help to train beginners and avoid the high rate of complications associated with the initial learning phase. An ex vivo porcine model for training in the transmural puncture and drainage of peripancreatic fluid collections has recently been developed, but its validity has not been proven [46]. A hybrid model (Mumbai EUS II) for stepwise teaching and training in EUS-BD and rendezvous procedures has also been developed [47]. Initial experience has shown that the model replicates situations encountered during rendezvous procedures and EUS-BD and that stepwise mentoring improves the chances of success in EUS-rendezvous and EUS-BD procedures. EUS-BD harbors several potentially life-threatening complications such as biliary peritonitis with sepsis and retroperitoneal bile fistula. Moreover, EUS-BD does not allow visual control of the color of the drained bile as with external drains, thereby potentially delaying the diagnosis of a major bleeding. Creating a broad fistula between the food containing stomach and the intrahepatic bile ducts is another subject of concern (sump syndrome). Overall, knowledge about long-term complications is limited.

## 4. Should EUS-BD and ERCP Be Performed by the Same Operator?

Standard basic gastroenterology and endoscopy training programs usually do not include advanced pancreatobiliary procedures, and therefore, a question arises as to whether an endoscopist undergoing specialized training in pancreatobiliary endoscopy should be trained in both EUS and ERCP.

### 4.1. Arguments in Favor 

The technique, interpretation and procedure all favor EUS being performed by an experienced ERCP endoscopist [48,49]. The technique of performing EUS in the duodenum with side-viewing instruments needs experience that can be achieved by training similar movements with duodenoscopes. The feeling for the passage into the descending duodenum, straightening of the endoscope and achieving a stable position without fluoroscopy are similar between EUS and ERCP. 

The interpretation of pancreatobiliary structures seen in EUS is better when one is familiar with the biliary anatomy and pathology from ERCP experience, and EUS findings can be directly correlated with the subsequent ERCP intervention. 

Unlike magnetic resonance imaging, EUS might be part of the real “one stop shop” for pancreatobiliary interventions if performed in the fluoroscopy room with the option to directly proceed to ERCP when necessary. If these advantages are already seen regarding diagnostic EUS, these become even more obvious when dealing with interventional EUS. Additionally, the ERCP endoscopist is already familiar with guidewire manipulation, dilatation, and stenting methods, which are prerequisites to EUS-guided interventions. Lastly, in case of the failure or complications of EUS, a high level of familiarity with the ERCP accessories and techniques is required to accomplish any rescue maneuver. 

### 4.2. Arguments Against 

The technique of maneuvering an instrument in the duodenum can be learned without experience in interventions at the papilla and beyond. Given the fact that only a small subset of gastroenterologists have enough technical experience to perform ERCP with a low associated complication level, teaching EUS cannot be restricted to those with high-level experience in ERCP. 

This is even more evident when one considers that the demand for diagnostic EUS is far greater than that of ERCP or interventional EUS. It may also be possible to train an ERCP endoscopist in EUS-BD techniques with focused training since ERCP endoscopists are familiar with wires, dilators, stents and side-viewing scopes so that they can have both options available to them while attempting an endoscopic drainage of the bile duct. The learning curve of EUS-BD has been reported to be approximately 100 appropriate interventions [50]. Moreover, it is not granted that the ERCP endoscopist learning just EUS-BD as a complement to ERCP is also competent in EUS pancreaticobiliary anatomy, biliary stone detection and tumor staging. 

On the other hand, ERCP endoscopists would benefit from training in EUS since this will allow combined diagnostic and therapeutic procedures, and even enable the exchange from one therapeutic procedure to another in the same session. 

## 5. Rendezvous Techniques 

Once the decision to proceed with EUS-BD has been made, multiple rendezvous approaches can be used. These consist either of antegrade stent placement or of retrograde stent placement. A rendezvous will only be attempted when the papilla is endoscopically accessible. First, the biliary system is punctured under EUS guidance. Subsequently, a guidewire is manipulated and advanced distally across the biliary stricture and then pushed through the papilla into the duodenum. Sometimes success is only achieved after inserting a cystotome that is helpful in performing difficult manipulation and avoiding the shearing of the guidewire. After the removal of the needle, the guidewire is left in place, the EUS scope is replaced using a duodenoscope, and the procedure is then completed with ERCP. 

EUS-rendezvous with ERCP was first reported in 2004 in two patients with malignant distal biliary obstruction in whom ERCP had been previously unsuccessful [4]. Since then, numerous other reports have been published in which the left-intra- or extrahepatic biliary ductal system were both used as access ports [51,52,53,54]. A systematic literature review on 20 studies [55] reported that EUS- rendezvous appeared safer than the transmural route with adverse events occurring in 11% of patients versus 21% for transmural drainage with EUS-HGS and EUS-CDS. This difference regarding adverse events was primarily related to avoiding bile leaks due to fistula formation during transmural drainage [56]. However, EUS-rendezvous has a lower success rate compared to transmural drainage [57], as recently reported in a review by Iwashita et al. [56], who found that EUS-rendezvous was technically successful in 81% of cases, which was significantly lower than those of EUS-HGS (87%) and EUS-CDS (94%). Most of the studies, however, were retrospective in nature, precluding any definitive conclusions. Nevertheless, all meta-analyses and systematic reviews published up to now have reported lower technical success rates as well as lower AE rates for EUS-rendezvous as compared with those of the other EUS-BD techniques [58,59]. 

### 5.1. Should Rendezvous Be Used First?

For benign diseases, EUS-rendezvous should be attempted first, and—until more data on the feasibility of more aggressive procedures in this clinical scenario are available—it should remain the only EUS interventional procedure attempted [60]. Because of its superior safety, EUS-rendezvous has also been used as a primary approach in malignant conditions. In consensus, EUS-rendezvous was the preferred technique to be initially used by most of the experts (32%) [18]. Importantly, from the beginning of the procedure, the endoscopist has to anticipate, in case of failure, to pass the guidewire into the duodenum; drainage can still be achieved using either an EUS-HGS or EUS-CDS approach, at least in most malignant etiologies [61]. 

An algorithm for EUS-BD for malignant CBD obstruction has been recently proposed, which is primarily based on the accessibility of the papilla [22,60]. If the papilla is accessible, EUS-rendezvous should be the preferred technique, while in cases of an inaccessible papilla, an alternative approach should be used based on the endoscopist’s preference and the institution’s available skills. Passing and manipulating the guidewire from the proximally dilated bile ducts through the malignant stricture and/or papilla into the duodenum may take a long time and can be more cumbersome than transmural stenting via EUS-CDS, especially after the introduction of an electrocautery-enhanced LAMS. Therefore, upfront EUS-CDS might be preferred over EUS-RV. Furthermore, the need to exchange the instrument and to then grasp and retract the guidewire inside the operative channel of the duodenoscope could be an argument in favor of a direct antegrade stenting through the echoendoscope, but this needs additional comparative evaluations. Importantly, if EUS-rendezvous fails, immediate drainage using an alternative technique (including PTBD) must be performed to avoid a biliary-peritoneal fistula with consecutive peritonitis and/or cholangitis.

### 5.2. Which Rendezvous Route Should Be Used?

With the intrahepatic route, the puncture of the biliary ducts is achieved by traversing the gastric wall and the liver parenchyma in segment 2 or 3. This is a difficult procedure for the following reasons: The more perpendicular the access route to the intrahepatic bile ducts is in relation to the longitudinal axis of the echoendoscope, the more cumbersome is the guidewire manipulation. Moreover, the movements of the liver with respiration and the smaller caliber of intrahepatic bile ducts make this approach very challenging. Nevertheless, in some cases (hilar stenosis surgically altered anatomy), it may be the only possible access. 

On the other hand, accessing the extrahepatic bile ducts has the advantage of the close proximity of the EUS probe within the duodenum, which has a thinner wall than the stomach, and of the relatively fixed retroperitoneal location of the CBD, with respiratory excursions. Access to the CBD can be performed either with the echoendoscope in the long position (from the duodenal bulb, with the needle directed toward the liver hilum) or in the short position (usually from the second duodenal portion—D2, with the needle directed to the ampulla). The latter is the preferred positioning for an EUS-rendezvous, but it is not always achievable. In a prospective small pilot study, guidewire passage to the duodenum was possible in 75% (3/4 cases), 100% (10/10 cases) and 60% (3/5 cases) of the EUS-RV attempts using the stomach, D2 and duodenal bulb approaches, respectively [60]. 

The rates of the adverse events of the intrahepatic or extrahepatic access routes for EUS-RV have been found to be similar. Importantly, there were two cases of pneumomediastinum among 11 patients in whom a trans-esophageal/trans-hepatic access route was chosen [62]. 

## 6. Percutaneous Transhepatic Cholangiography and Biliary Drainage (PTBD)

PTBD and surgical bypass have traditionally been the salvage procedures in cases of failed ERCP. In patients with biliary dilatation, the PTBD approach is successful in 86%, while it is only technically feasible in 63% without dilated bile ducts according to guidelines by American radiologists [63]. In experienced hands, in dilated bile ducts, the success rate is close to 100%. In patients with hilar carcinoma types III and IV, PTBD seems to have lower rates of cholangitis and acute pancreatitis than stenting via ERCP [64]. 

After failed ERCP for distal malignant obstruction, EUS-BD outperforms PTBD as a salvage approach as it has a decreased risk of stent or catheter dysfunction, requiring further reintervention (RR, 0.37; 95%CI, 0.22–0.61), and adverse events (RR, 0.59; 95%CI, 0.39–0.87), as demonstrated in a meta-analysis including three randomized trials [65]. Stent patency is, however, an important factor in these patients, as dysfunction will negatively impact neoadjuvant or palliative chemotherapy. The risk of seeding metastasis is often not taken into account when choosing the biliary drainage method. PTBD has a significantly higher risk of causing seeding metastasis than endoscopic biliary drainage [66] (22.0% vs. 10.5%, respectively; *p* < 0.00001).

### 6.1. Arguments in Favor of PTBD

PTBD can lead to a rapid and effective resolution of cholangitis. It requires an X-ray workstation, even if the access is sought percutaneously through ultrasound in modern centers. Access is not always easy and can take some time, especially if the bile ducts are not very wide intrahepatically. Sufficient sedation, e.g., propofol sedation, is required. 

In specialized centers, it is usually possible to create an external–internal drainage with PTBD. If it is not possible to internalize the drainage immediately, it is recommended to drain the bile externally first. After one or two days, when the bile ducts are less dilated, the wire is no longer looped in the dilated bile duct segment and can usually be advanced internally through the stenosis. Whether to implant a metal stent or perform external–internal Yamakawa drainage requires discussion with the patient. To avoid subsequent interventions, a metal stent is usually implanted in tumor stenoses. However, it must be remembered that external–internal drains preserve access in the event of tumor progression.

Despite some disadvantages, PTBD remains an important modality in the toolbox of interventional gastroenterologists and radiologists. Furthermore, the literature data about risks and benefits are far more robust than those regarding novel EUS-guided approaches.

Finally, not all anatomical situations are suitable for endoscopic drainage, e.g., a right hepatic biliary tree dilation, separated by a hilar or intrahepatic stenosis, still represents a rather elective indication for PTBD in case of failed ERCP.

### 6.2. Arguments Against

The disadvantage of PTBD is that in patients with insurmountable stenosis, it may fail to establish internal drainage, determining the need for a life-long external catheter. The reported higher rate of adverse events could be due to selection bias, and the reported higher rate of pain could be diminished if performed under the same sedation depth as endoscopic procedures. The pain in PTBD depends on the sedation, on the length of the intervention with biliary peritonitis appearing if bile fluid enters the peritoneal cavity and/or on the technical skill of the operator. After ERCP failure for a stent distal malignant obstruction, EUS-BD is superior to PTBD as a salvage procedure due to less stent dysfunction requiring less reintervention (RR, 0.37; 95%CI, 0.22–0.61) and less adverse events (RR, 0.59; 95%CI, 0.39–0.87) [65,67]. However, proper stent function is an important factor for these often-frail patients as it will affect the oncological outcomes of neoadjuvant or palliative chemotherapy. Nevertheless, it is difficult to compare different methods properly due to the multifaceted complexity of such techniques. 

### 6.3. Should PTBD and EUS-BD Be Performed by the Same Physician?

In the United States, PTBD is performed by interventional radiologists, and EUS-BD, by gastroenterologists who are trained in EUS. There is no crossover, but this is different in other parts of the world. It could be advantageous to offer both methods of biliary drainage with a high success rate in every institution. This is of importance since both methods are not mutually exclusive. In addition, PTBD offers the option to perform cholangioscopy (traditional and single-use cholangioscopes, sometimes also bronchoscopes). EUS-BD offers the opportunity to perform direct stenting and bilioenterostomy in case of unsurmountable stenosis. It is doubtful, though, whether the same examiner can ever achieve comparably high skills for both methods in most institutions. In addition, it does not really matter who performs what but that dedicated interventions can be performed in the individual treatment center. In some health care systems, where specialized training is compartmentalized based on specialty, only radiologists are formally trained in PTBD as an interventional radiology technique. Similar to EUS and ERCP, whoever performs PTBD should be properly trained in the procedure regardless of which department they are a part of. 

## 7. Do We Need Cystotomes?

EUS-BD is a multi-step process. Creating a fistula between the gut and the bile duct lumen is one of the most challenging steps. After the puncture and insertion of a guidewire the tract must be enlarged to allow stent insertion. However, with lumen-apposing stents, this step is usually not necessary. Novel advances in stent design aim to reduce the caliber of the stent delivery systems further such that tract dilatation can be completely avoided [68]. 

Two established techniques for tract formation and enlargement are mechanical dilation and electrocautery. In the former, the tract is enlarged mechanically using conventional bougies or a 4 mm dilating balloon as used in ERCP [7,47,69,70,71]. Although dilating balloons are very efficient for bile duct stenosis, this is not the case for EUS-guided tract dilation [72,73], such as EUS-HGS where the tract is long; the passage through the proper muscle layer, difficult; and the position of the echoendoscope in the stomach, unstable. In this situation, it is difficult to optimize the axis to advance the balloon, even over a guidewire. Furthermore, the elasticity of the gastric wall and the resistance of fibrotic tissue (bile duct, liver) hinder the advancement of dilation devices. Bougienage and electrocautery are better options. The success of bougienage alone is not consistent, and electrocautery devices may be required. In a trial by Park et al., graded dilation alone was successful in only 74% of cases for EUS-HGS (74%) and 21% for EUS-CDS (21%) [7]. 

Two electrocautery devices have been used, non-coaxial (needle knife) and coaxial (cystotome, Will-knife^®^). The problem of a catheter deployed at these locations is the angulation of the echoendoscope. Even over a guidewire, the needle-knife may point tangentially, increasing the risk of an inadvertent incision and subsequent pneumoperitoneum or bleeding [7,74,75,76]. The use of a needle-knife has been shown to be associated with more post-procedure adverse events compared to graded dilation alone (9/27, 33% vs. 2/28, 7%; *p* = 0.02) [7].

Cystotomes, also called diathermic dilators, were first applied in pseudocyst drainage via gastric or duodenal routes [77]. A 10 Fr cystotome, including a needle-knife and outer catheter with a diathermic tip, is routinely used for this purpose using a brief burst of pure cutting current. These devices have the advantage of being coaxial [72]. As 10 Fr cystotomes are too large to be used in the liver, the industry designed a smaller 6 Fr cystotome with a diathermic tip and no needle-knife. Although the efficiency of this device is established, the cautery dilation may cause an acute and late “burn effect” to the vessels around the needle tract, as well as bleeding [7,70,78]. In a recent study comparing a new mode of an ultra-tapered mechanical dilator with a coaxial cystotome, the efficacy was similar, but bleeding was observed significantly more frequently (18% vs. 0%, *p* = 0.04) in the cystotome group [78]. The impact of such bleeding is usually very low, as fully covered stents are inserted, which produce a hemostatic effect by compressing the vessels. Nevertheless, an alternative method should be favored for patients with a bleeding tendency or when interposing blood vessels cannot be avoided. 

Different companies produce 6 Fr cystotomes that are available in some countries (cystogastroset (6; 8.5; 10 Fr single use), Endoflex, Voerde, Germany/cystogastrostome, (6, 8.5, 10 Fr reusable or single use), G-flex, Nivelle, Belgium/ cystotome (6, 7, 8, 10 Fr single use) Shaili endoscopy, Gujarat, India/ ring knife Prof. Dr. U. Will (1.3 mm single use) MTW, Wesel, Germany). There are currently no trials comparing the different cystotomes, but the possibility to inject contrast medium during the procedure without removing the guidewire is an advantage of the G-flex cystotome.

## 8. Do We Need Bougies?

Enlarging the needle tract is an important step before the introduction and deployment of biliary stents. The use of mechanical dilation may reduce the risk of damaging the surrounding structures. As mentioned above, widening the fistula using an electric cautery device is technically easy but has the risks of bleeding, perforation, incorrect puncture path and burn injury. Mechanical dilatation instruments include bougies, dilatation balloons and stent retrievers. However, inserting such dilatation instruments [79] over the guidewire is often difficult. During bougie exchange, a loop formation of the wire can occur, especially when correcting the wire position. Exchanging instruments poses the risk of losing the position and access to the biliary system. The excessive dilatation of the fistula tract might result in postponed bile leakages; hence, maximal dilatator sizes should be carefully selected according to the planned stent caliber and length.

Fine-tip tapered balloon catheters are available, which are easily inserted after a 19 G needle puncture without a previous dilatation of the new fistula tract. Balloon dilatation ensures adequate radial force for tract dilatation and avoids sequential steps of instrument changes for the different-sized bougies over the wire. 

## 9. Are All EUS Needles the Same for EUS-BD?

In most instances, a 19 G needle is used to gain access to the biliary system; 19 G needles allow the introduction of 0.035 guidewires to stabilize the position. Using smaller needles would only allow the insertion of smaller and less stable wires that would not enable the safe deployment of stents into the bile ducts but only facilitate rendezvous techniques.

Many standard 19 G aspiration needles can be used to puncture the targeted bile duct under EUS guidance. 

A specially designed 19 G access needle is available that cuts with a sharp stylet tip (EchoTip^®^ Ultra HD Access Needle, Cook Medical, Bloomington, IL, USA). Therefore, the stylet must be fully inserted during the puncture; otherwise, the needle tip would be blunt. When the stylet is not in place, the edges of the needle are blunt (in contrast to conventional needles), and it is safe to pull the wire back or correct the wire position without the risk of shearing. However, some investigators have experienced that shearing the wire is also possible with the blunt needle in cases with very sharp bends. A flexible nitinol-covered 19 G needle is easier in maneuvering, especially in a bended position of the echoscope, but head-to-head comparisons with conventional needles are lacking. Recently, an access system has been introduced (Beacon EUS Access System; Medtronic Inc., Minneapolis, MN, USA) that punctures the bile duct whilst the sharp stylet is fully inserted. The removal of the stylet causes the blunt, tipped 18.5 G access catheter to bend to a pre-determined 90 or 135 degrees. The access catheter is also fully rotatable, which facilitates selective wire advancement and direction [80]. Blunt edges that avoid the shearing of the wire and anchoring due to the angulation also seem helpful in this new design.

## 10. Plastic or Metal Stents?

In a systematic review including 42 studies and 1192 patients, the technical success rate and the clinical success rate were 94.5% and 92.5%, respectively, for all the EUS-BD techniques [58]. The clinical success rate of studies using plastic stents versus those using metal stents was not statistically different (98.2% vs. 94.5%, respectively). However, in the same large systematic review, the rate of adverse events associated with metal stents was significantly lower when compared to that of plastic stents (17.5% vs. 31%) (*p* = 0.013) [58]. Similar conclusions have been drawn in a large multicenter retrospective study on 240 patients who underwent EUS-BD, in which plastic stents were associated with a higher incidence of cholangitis when compared to the fully covered SEMSs (11% vs. 3%; *p* = 0.02), while bile leak rates were similar between the two groups (9.3% vs. 9.2%; *p* = 0.97) [10]. As a result, plastic stents have been abandoned for these procedures in favor of the fully covered SEMSs in many centers. 

A randomized multicenter study compared stent patency and complication rates between plastic stents and SEMSs in patients with unresectable, distal malignant biliary obstruction. In the plastic stent group, the frequency of stent failure was significantly higher compared with the SEMS group [81]. 

This is also true when dealing with LAMS used under EUS-guidance, which are able to create a real and stable anastomosis between the gastrointestinal lumen and the biliary system or the gallbladder [26,27,82,83,84]. Although the larger diameter of metal stents reduces the risk of stent occlusion, preliminary evidence suggests that a plastic stent be placed through the LAMS to maintain the orientation of the axis and reduce the need for reintervention [85]. 

Recently, an ESGE technical review issued a strong recommendation for partially or fully covered self-expandable metal stents or small caliber lumen-apposing metal stents, but with a moderate quality evidence. On the other hand, plastic stents were not recommended by the ESGE review for biliary drainage [29]. 

## 11. Adverse Events 

A recent meta-analysis pooled the adverse events rates of 7887 EUS-BD procedures from 55 studies, reporting an overall 13.7% rate with major adverse events and mortality occurring in only 0.6% and 0.1% of cases, respectively. Bile leaks (2.2%) and cholangitis (1%) were the most common early adverse events, while reinterventions due to stent occlusion or stent migration were performed in 16.2% of cases [67]. 

Overall, 24 studies were included in the final analysis. Technical success was comparable between EUS-BD and PTBD (OR = 1.12, 0.67–1.88). However, EUS-BD was associated with a higher clinical success rate (OR = 2.55, 1.63–4.56) and lower adverse events (OR = 0.41, 0.29–0.59) compared with PTBD. While the incidence of major adverse events (OR = 0.66, 0.31–1.42) and procedure-related mortality (OR = 0.43, 0.17–1.11) were similar between the two groups, EUS-BD was associated with lower chances of reintervention (OR: 0.20, 0.10–0.38). 

In a retrospective multicenter study on 240 patients undergoing intrahepatic and extrahepatic EUS-BD for benign and malignant conditions, the most important adverse events for all the techniques were bleeding (11%), bile leak/peritonitis (10%), cholangitis (5%) and pneumoperitoneum (5%) [10]. These adverse events were independent from the access route (32.6% with intrahepatic vs. 35.6% extrahepatic with an extrahepatic approach; *p* = 0.64) and from the nature of the obstruction (26.7% with benign vs. 37.1% with malignant nature; *p* = 0.19) [10]. The most common complication in 103 patients with EUS-BD and benign cholestasis was stent dislocation during EUS-hepaticogastrostomy [86]. More details regarding adverse events are shown in Table 1.

## 12. Conclusions

EUS-BD and EUS-GBD have become important and effective minimally invasive and safe modalities for treating biliary obstruction over the past years. EUS-BD can be performed when ERCP fails, or as an alternative to percutaneous drainage or when a surgical procedure is considered to put the patient at high risk. EUS-BD as a primary method of biliary drainage has been proposed for patient care as well and not just as an alternative when conventional decompression methods fail. The controversies of EUS-BD and EUS-GBD are discussed in detail in this comprehensive paper.

## Figures and Tables

**Figure 1 cancers-16-01616-f001:**
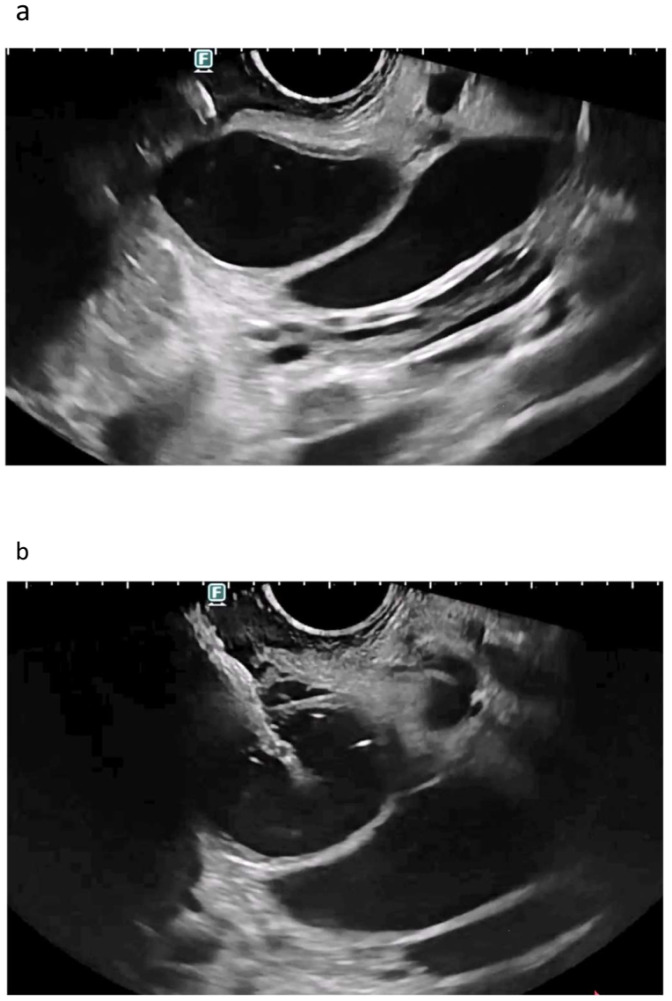
EUS-CDS misdeployment. Anatomical delineation of the common bile duct next to the tip of the instrument (**a**). Tip of the electrocautery-enhanced lumen-apposing metal stent before deployment (**a**). Distal flange of the LAMS opened transmurally between the bile duct and the duodenal wall (**b**). Despite the flange being opened transmurally, the tip of the device was correctly inside the bile duct, and therefore, a guidewire was moved toward the hilum (**c**). The LAMS was recaptured and moved over the wire for a fluoroscopy-guided release, with the correct placement of the distal flange inside the bile duct (**d**). Final placement of the LAMS with the distal flange inside the bile duct and the proximal flange in the duodenum (left: endosonography; right: fluoroscopy confirming aerobilia through the LAMS (**e**).

**Figure 2 cancers-16-01616-f002:**
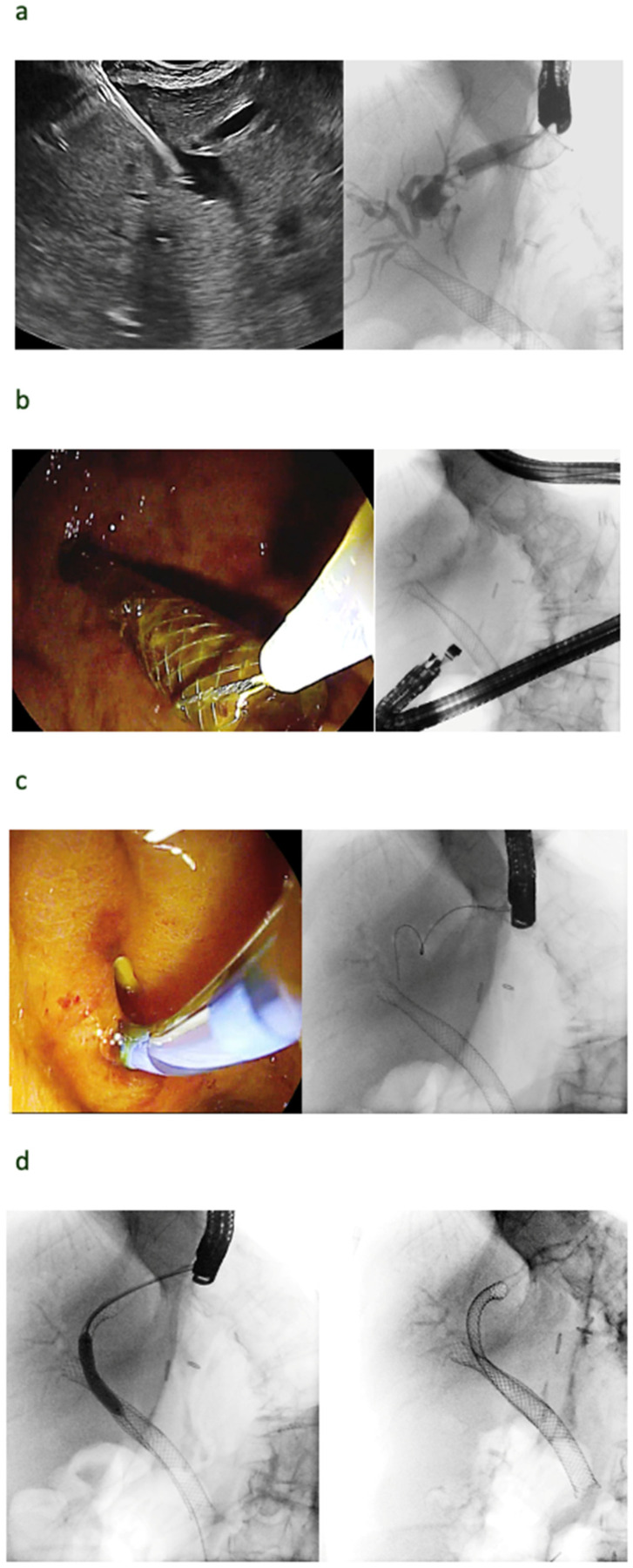
EUS-HGS dislodgement. A 75 years-old lady with metastatic pancreatic cancer and a hilar metastasis underwent EUS-guided hepaticogastrostomy for the drainage of the left liver lobe (**a**). At a second endoscopic procedure 10 days later, meant for the optimization of biliary drainage on the right, a dislodgement of the former EUS-HGS stent was caused. The stent was, therefore, removed with a snare (**b**). The mature gastrohepatic fistula was identified and cannulated with a sphincterotome over the wire (**c**). The guidewire was redirected toward the hilum, through the former biliary stent, and, after dilation, this guidewire was used for the antegrade placement of an uncovered metal stent for the drainage of the left liver lobe (**d**).

**Figure 3 cancers-16-01616-f003:**
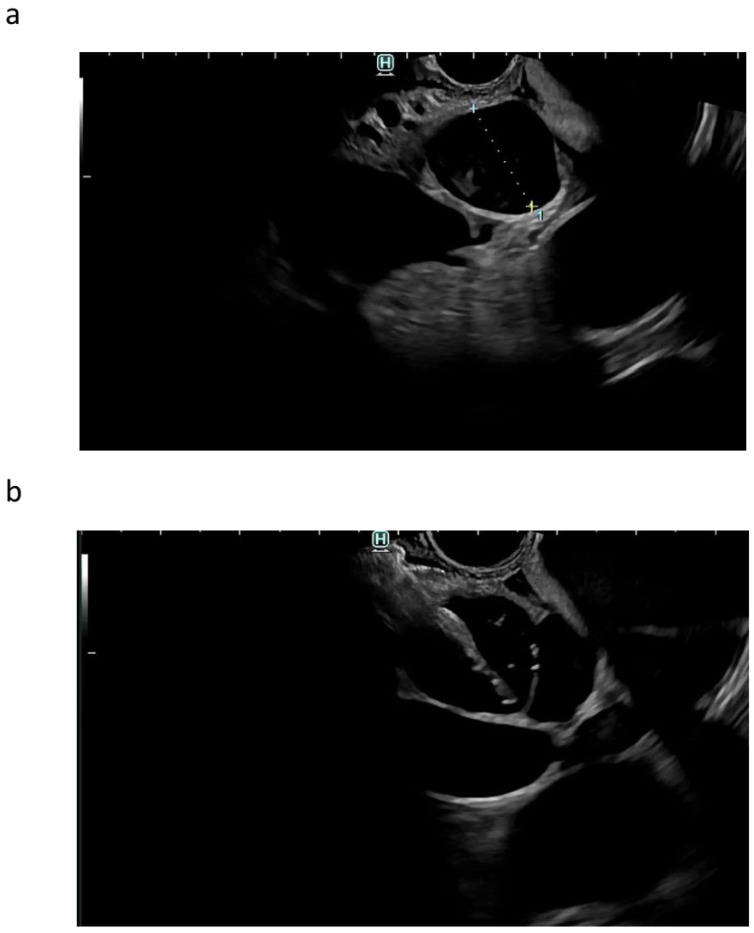
EUS-guided choledochoduodenostomy. EUS-guided identification of the dilated common bile duct above the neoplasia from the bulb (**a**). EUS-guided penetration of the common bile duct with the electrocautery-enhanced lumen-apposing metal stent (**b**). Release of the distal flange inside the common bile duct, and traction in preparation for the intrachannel release of the proximal flange (**c**). EUS appearance of the released stent (air [CO_2_] flowing inside the stent) (**d**). Endoscopic appearance of the stent, draining bile (**e**). Radiologic appearance of the stent, with aerobilia depicting the biliary tree (**f**).

**Figure 4 cancers-16-01616-f004:**
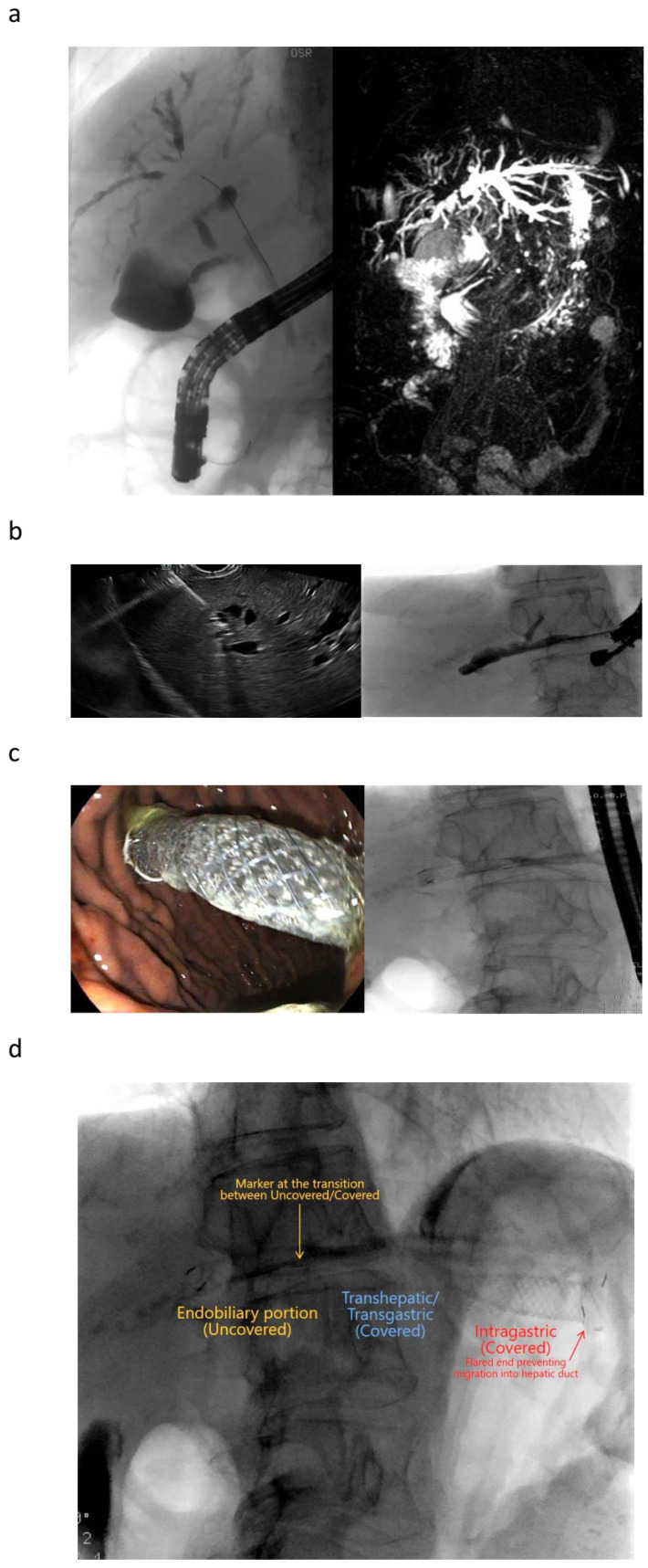
EUS-guided hepaticogastrostomy. Patient with pancreatic cancer and liver metastasis, one of which is obstructing the biliary hilum and separating the biliary hemisystems. (**a**) LEFT: Retrograde cholangiography obtaining only cannulation to the right. RIGHT: MRI-cholangiography showing a massive dilation of the left biliary tree. (**b**) LEFT: Identification of a segment 2 bile duct with a course from the top-left to the bottom-right of the screen (i.e., from the direction of the operative channel of the endoscope to the liver hilum). Transgastric puncture of the duct with a 19 G needle. RIGHT: Contrast injection after bile duct aspiration. (**c**) LEFT: Endoscopic view of the intragastric (covered) portion of the stent. RIGHT: Radiological appearance of the released stent. (**d**) Release of the uncovered portion of the stent inside the biliary tree, with the covered portion transverses the liver parenchyma and the gastric wall ending with a long portion inside the gastric lumen. (**e**) Restaging CT after 15 days, demonstrating a complete resolution of the biliary dilation to the left. Bilirubin was dropped from 17 to 2.2 mg/dL.

**Table 1 cancers-16-01616-t001:** Adverse events in EUS-guided bile duct drainage.

Autor	Study Type	Patients	Adverse Events	Comments
Burmester et al. Gastrointest Endosc 2003 [3]	Single-center case reports	4	Bile leak (*n* = 1)	EUS-CDS, retrograde HJS, EUS-HGS.
Puspok et al. Am J Gastroenterol 2005 [87]	Case reports	6	Cholecystitis (*n* = 1)	EUS-guided transduodenal puncture of the common bile duct with stent placement. Transhepatic metal stent
Kahaleh et al. Gastrointest Endosc 2006 [88]	Retrospective study	28	Minor bleeding (*n* = 1), self-limited Pneumoperitoneum (*n* = 2) Bile leak (*n* = 1)	Rendezvous
Bories et al. Endoscopy 2007 [89]	Pilot study, single-center	11	Cholangitis (*n* = 1)	EUS-HGS
Yamao et al. Endoscopy 2008 [90]	Case reports	5	Pneumoperitoneum (*n* = 1)	EUS-CDS
Tarantino et al. Endoscopy 2008 [51]	Single-center	9	Death from liver cirrhosis complication 15 days after intervention (*n* = 1)	Transduodenal approach
Maranki et al. Endoscopy 2009 [52]	Single-center, retrospective	49	Pneumoperitoneum (*n* = 3), Bleeding (*n* = 1), Aspiration pneumonia (*n* = 1)	Transgastric-transhepatic (intrahepatic) or transenteric-transcholedochal (extrahepatic)
Horaguchi et al. Dig Endosc 2009 [91]	Single-center	16	Peritonitis (*n* = 1)	EUS-BD via duodenum, stomach, esophagus
Brauer et al. Gastrointest Endosc 2009 [92]	Comparative study, single-center nonrandomized observational study	20	Pneumoperitoneum (*n* = 1) Respiratory failure (*n* = 1)	Transenteric-transcholedochal extrahepatic approach for biliary cases
Park et al. Gastrointest Endosc 2011 [7]	Comparative study, prospective follow-up study.	57	Postoperative adverse effects after EUS-EUS-BD: 20%: bile peritonitis (*n* = 2), mild bleeding (*n* = 2), and self-limited pneumoperitoneum (*n* = 7) Late adverse effecta: distal stent migration—7%.	In multivariate analysis, needle-knife use was the single risk factor for post-procedure adverse events after EUS-BD
Hara et al. Am J Gastroenterol 2011 [93]	Prospective study	18	Peritonitis (*n* = 2) Bleeding (1)	EUS-CDS
Artifon et al. J Clin Gastroenterol 2012 [13]	Randomized controlled trial	25	Bile leak (*n* = 1) Mild bleeding (*n* = 1)	Percutaneous transhepatic biliary drainage and EUS-BD
Kim et al. World J Gastroenterol 2012 [94]	Two-center study	13	Peritonitis (*n* = 1)	EUS-CDS and EUS- HGS, fully nitinol-covered self-expandable metal stent
Gupta et al. J Clin Gastroenterol 2014 [10]	Multicenter, nonrandomized retrospective study.	240	Pneumoperitoneum 5%, bleeding 11%, bile leak/peritonitis 10% and cholangitis 5%.	Extra- and intrahepatic BD access. No significant difference between IH and EH approaches; benign and malignant indications
Will, et al. Ultraschall Med 2015 [8]	Single-center database over a 10-year period	95	Elevated cholestasis parameters (*n* = 3)	EUS-BD
Vanella et al. EIO 2020 [95]	Retrospective	104	Perforation (*n* = 2), Bleeding (*n* = 3), Bile leak (*n* = 1), Cholangitis (*n* = 9), Bacteriemia (*n* = 3), Acute Pancreatitis (*n* = 4), Severe abdominal pain (*n* = 2)	EUS-guided intrahepatic access, including rendezvous, antegrade stenting, and EUS-HGS
Füldner et al. Z Gastroenterol 2021 [86]	Prospective EUS-BD registry (2004–2020)	103	Complication rate: (*n* = 26/25%): stent dislocation (*n* = 11), perforation (*n* = 1), pain (*n* = 2), hemorrhage (*n* = 6), biliary ascites/leakage (*n* = 3) and bilioma/liver abscess (*n* = 3); major complication rate (*n* = 12/68—17.6%).	Different approaches of EUS-BD
Venkatachalapathy et al. Gastrointest Endosc 2021 [96]	Prospective multicenter study	20	Cholangitis (*n* = 1/5%), Stent migration (*n* = 1/5%)	EUS-CDS, lumen-apposing metal stents
Marx et al. Endosc Ultrasound 2022 [97]	RCT	35	Bleeding (*n* = 3/8.6%) Cholangitis (*n* = 1/2.9%) Peritonitis (*n* = 1/2.9%) Sepsis (*n* = 7/20%)	EUS-HGS
Ragab et al. Acta Gastroenterol Belg 2023 [98]	Prospective multicenter study	91	AE rate 18.7% for EUS-HGS, AE rate 8.9% for EUS-CDS. AE not specified.	EUS-HGS (*n* = 35) EUS-CDS (*n* = 48)
Bun Theo et al. Gastroenterology 2023 [28]	RCT	83	Cholangitis (*n* = 5) Stent misdeployment (*n* = 2) Stent migration (*n* = 1) Multi-organ failure (*n* = 2) Fatal (*n* = 4)	EUS-CDS with Hot Axios

EUS-BD: endoscopic ultrasound-guided biliary drainage; CDS: choledocho-duodenostomy; HGS: hepatico-gastrostomy; HJS: hepaticojejunostomy.

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
