# Peer review of "Controversies in Endoscopic Ultrasound-Guided Biliary Drainage"

_cancers, 2024, doi:10.3390/cancers16091616_

Round 1
Reviewer 1 Report
Comments and Suggestions for Authors
This was a very well-done study and the data reported is succinct and well-presented. After carefully review this paper, I think this paper was still need further revision and add some detail information for future decision.
1. In figure 2, please re-arrangement figure 2b and 2c. The endoscopy picture was in left side and radiologic picture in right side.
2. In “Arguments against EUS-guided drainage” part
According to a previous study, the most common concomitant chronic dis[1]ease in elderly adults who undergo ERCP is cardiovascular disease (44.6–57.6%), with concomitant cerebrovascular disease rates of approximately 8% to 18.2% in elderly adults. Many of these individuals are treated with long[1]term antiplatelet or antithrombotic agents.
The EUS-guided drainage was also high bleeding procedure. Authors might consider addressed the possibility of bleeding risk of EUS-guided drainage when them had accepted antiplatelet or antithrombotic agents treatment.
Author Response
please see attachement

Reviewer 2 Report
Comments and Suggestions for Authors
l This manuscript performed a valuable review about the evolving aspect of biliary drainage techniques, emphasizing the usefulness of EUS-guided approaches as alternatives to traditional methods.
l This review also provides a lot of comprehensive information that would be helpful for hepato-biliary physician when dealing with the complexities and controversies about EUS-guided biliary drainage. It includes a balanced overview of the related advantages, limitations, and areas to be further investigated.
Author Response
We very much thank and appreciate the very kind review.
Reviewer 3 Report
Comments and Suggestions for Authors
Author Response

(The authors gave the same response as above.)

Reviewer 4 Report
Comments and Suggestions for Authors
The article is a literature review. The authors discuss the advantages and disadvantages of modern high-tech methods of drainage of the biliary tract in patients with biliary stenosis of various etiologies. The authors give recommendations on the use of these technologies in a really clinical situation. The article is recommended for publication. The paper can be accepted without any further changes.
Author Response

(The authors gave the same response as above.)

Round 2
Reviewer 1 Report
Comments and Suggestions for Authors
This was a very well done study and the data reported is succinct and well-presented. Authors had complete response the previous reviewers’ suggestion and revised well. May consider accepted this article.